



# flat10MIP: An emissions-driven experiment to diagnose the climate response to positive, zero, and negative CO2 emissions

Benjamin M. Sanderson[1], Victor Brovkin[2], Rosie A. Fisher[1], David Hohn[3], Tatiana Ilyina[4,5,2], Chris D. Jones[6,7], Torben Koenigk[8], Charles Koven[9], Hongmei Li[5,2], David M. Lawrence[10], Peter Lawrence[10], Spencer Liddicoat[6], Andrew H. MacDougall[11], Nadine Mengis[3], Zebedee Nicholls[12,13,14], Eleanor O'Rourke[15], Anastasia Romanou[16,17], Marit Sandstad[1], Jörg Schwinger[18], Roland Séférian[19], Lori Sentman[20], Isla R. Simpson[10], Chris Smith[13,21], Norman J. Steinert[1], Abigail L. S. Swann[22], Jerry Tjiputra[18], Tilo Ziehn[23]

1.    CICERO International Center for Climate Research, Oslo, Norway
2.    Max Planck Institute for Meteorology, Hamburg, Germany
3.    GEOMAR, Helmholtz Centre for Ocean Research, Kiel, Germany
4.    University of Hamburg, Hamburg, Germany
5.    Helmholtz-Zentrum Hereon, Geesthacht, Germany
6.    Met Office Hadley Centre, Exeter, UK
7.    School of Geographical Sciences, University of Bristol, Bristol, UK
8.    Swedish Meteorological and Hydrological Institute (SMHI), Norrköping, Sweden
9.    Lawrence Berkeley National Laboratory, Berkeley, CA, USA
10.   NSF National Center for Atmospheric Research (NCAR), Boulder, CO, USA
11.   St. Francis Xavier University, Antigonish, NS, Canada
12.   University of Melbourne, Melbourne, Australia
13.   International Institute for Applied Systems Analysis, Austria
14.   Climate Resource, Fitzroy, Australia
15.   CMIP Project Office
16.   NASA Goddard Institute for Space Studies, New York, NY, USA
17.   Columbia University, New York, USA
18.   NORCE Norwegian Research Centre, Bjerknes Centre for Climate Research, Bergen, Norway
19.   Centre National de Recherches Météorologiques (CNRM), Toulouse, France
20.   NOAA Geophysical Fluid Dynamics Laboratory (GFDL), Princeton, NJ, USA
21.   University of Leeds, Leeds, UK
22.   University of Washington, Seattle, WA, USA
23.   CSIRO Environment, Aspendale, Australia

*Correspondence to*: Benjamin M. Sanderson (benjamin.sanderson@cicero.oslo.no)

**Abstract.** The proportionality between global mean temperature and cumulative emissions of $CO_2$ predicted in Earth System Models (ESMs) is the foundation of carbon budgeting frameworks. Deviations from this behavior could impact estimates of required net zero timings and negative emissions requirements to meet the Paris Agreement climate targets. However, existing ESM diagnostic experiments do not allow for direct estimation of these deviations as a function of defined emissions pathways. Here we perform a set of climate model diagnostic experiments for the assessment of Transient Climate Response to cumulative $CO_2$ Emissions (TCRE), Zero Emissions Commitment (ZEC), and climate reversibility metrics in an emissions-driven framework. The emissions-driven experiments provide consistent independent variables simplifying simulation,



analysis and interpretation with emissions rates more comparable to recent levels than existing protocols using model-specific compatible emissions from the CMIP DECK *1pctCO2* experiment, where emissions are strongly weighted towards the end of the experiment at significantly greater than present day values. A base experiment, '*esm-flat10*', has constant emissions of $CO_2$ of 10GtC per year (near-present day values), and initial results show that TCRE estimated in this experiment is about 0.1K less than that obtained using 1pctCO2. A subset of ESMs exhibit land carbon sinks which saturate during this experiment. A branch experiment, *esm-flat10-zec*, measures ZEC, which we find is reduced by 25-50% compared with 1pctCO2 branch experiments. A final experiment, *esm-flat10-cdr*, assesses climate reversibility under negative emissions, where we find that peak warming may occur before or after net zero and residual warming after removal of all greenhouse gases is well described by ZEC in most models and that current Simple Climate Model (SCM) distributions may be over-estimate temperature reversibility compared with ESMs. We propose a set of climate diagnostic indicators to quantify various aspects of climate reversibility. These experiments were suggested as potential candidates in CMIP7 and have since been adopted as "fast track" simulations.

## 1. **Introduction**

The concept of proportionality of global mean temperatures to cumulative carbon dioxide emissions is central to carbon budgeting frameworks and net zero commitments(Rogelj et al., 2019b). The relationship has its origins in the recognition of a robust linear relationship in Earth System Model simulations between the global mean temperature change and the cumulative amount of $CO_2$ released into the atmosphere, the slope of which we refer to as the Transient Climate Response to Cumulative Emissions (TCRE) - the change in global mean temperature per trillion tonnes of carbon emitted to the atmosphere (Allen et al., 2009; Matthews et al., 2009; Zickfeld et al., 2009). TCRE offers a powerful, simplified lens for climate policy applications, allowing policymakers to directly equate emission budgets to projected warming levels (Lamboll et al., 2023; Rogelj et al., 2019b), and to gauge the relative impact of different emissions trajectories over time (MacDougall, 2015).

For a simulation in which temperature changes are driven by $CO_2$ alone,

$$TCRE = \frac{\Delta T(t)}{I_{em}(t)}, \tag{1}$$

where $\Delta T(t)$ and $I_{em}(t)$ are the temperature change and cumulative emissions at time *t, respectively*.

In order to apply this approach to constrain compatible carbon emissions budgets, a correction must first be made for the temperature impact of present and future non-$CO_2$ emissions (Rogelj et al., 2019b). Multiple approaches have been proposed, either by assuming a ratio of future $CO_2$ and non-$CO_2$ emissions (Damon Matthews et al., 2021; Leach et al., 2018; Millar and Friedlingstein, 2018), by subtracting an estimate of non-$CO_2$ warming (Lamboll et al., 2023) or by defining a TCRE based on cumulative $CO_2$- forcing-equivalent emissions (Jenkins et al., 2021).



Robust use of TCRE to assess compatible emissions also requires an assessment of the limits of its applicability. For climate
models, TCRE is generally calculated using results from a concentration-driven simulation *1pctCO2*, where $CO_2$
concentrations are prescribed and ramped up exponentially at a rate of 1% per year. In assessments (Intergovernmental Panel
on Climate Change, 2023), the TCRE nominally computed in year 70, when concentrations have approximately doubled:

$$TCRE_{1pctCO2} = \frac{\Delta T(70)}{I_{em}(70)} = \left(\frac{\Delta T(70)}{I_{atmos}(70)}\right)\left(\frac{I_{atmos}(70)}{I_{em}(70)}\right),$$

where $I_{atmos}(70)$ is the additional carbon in the atmosphere and $\Delta T(70)$ is the Transient Climate Response (TCR, in practice
calculated as the average of years 60-80). $\frac{I_{atmos}(70)}{I_{em}(70)}$ is the cumulative airborne fraction, the proportion of cumulative emissions
which remain in the atmosphere.

The potential for deviations from cumulative emissions proportionality was summarized in the Zero Emissions Commitment
(Intergovernmental Panel on Climate Change, 2023; Lamboll et al., 2023) where it was used as an estimate of additional
temperature change which could be expected after emissions have reached net zero, alternatively framed as a 'Rate of
Adjustment to Zero Emissions' or RAZE which is used to define an emissions rate which would be consistent with constant
temperatures (Jenkins et al., 2022).

Our definitions of ZEC are, to date, primarily informed by the ZECMIP CMIP6 experiment (Jones et al., 2019) which is based
on an abrupt cessation of emissions branching from the *1pctCO2* experiment when 1000PgC of $CO_2$ emissions have been
diagnosed, where ZEC50 and ZEC100 is the temperature change 50 and 100 years respectively after the cessation of emissions.
This experiment was performed by a coordinated set of Earth System Models and intermediate complexity models, which led
to the finding that ZEC had the potential to be either positive or negative (MacDougall et al., 2020) with a best estimate near
zero.

ZECMIP experiments were designed this way to ensure consistency of ZEC and TCRE at the same point, but they do, however,
have a number of limitations. Firstly, 1pctCO2 is a prescribed concentration trajectory for atmospheric $CO_2$, and compatible
emissions are computed as a residual term, such that each climate model has a different emissions trajectory. This poses two
issues for using the run as a basis for the assessment of ZEC. Firstly, each model follows its own pathway of (implied)
emissions in such experiments, obfuscating the relationship between model and ZEC response. Secondly, the compatible
emissions profile in 1pctCO2 grows throughout the experiment, with the burden of cumulative emissions weighted towards
the end of the experiment ((Sanderson et al., 2023) and Figure 6), whereas contemporary emissions are closer to flat (since
about 2012).




In addition, no experiment within prior CMIP efforts has been designed to robustly understand the degree of asymmetry in the climate response to positive followed by negative $CO_2$ emissions. The compatible emissions from the *1pctCO2-cdr* concentration reversal experiment used in CDRMIP (Asaadi et al., 2024)are both asymmetric in time, between the positive and negative emissions periods, and have a large discontinuity of roughly 50 Pg C/yr (Koven et al., 2023) at the point of

reversal from increasing to decreasing $CO_2$ concentrations. This large discontinuity in emissions causes transient temperature responses (Zickfeld et al., 2016) which inhibit a clear diagnosis of whether and how the general climate response to negative emissions differs from the climate response to positive emissions (MacDougall, 2019). An idealized CMIP experiment that allows for a continuous transition from positive to negative emissions, and one that is symmetric in time (so that any asymmetries that arise are driven by the coupled carbon-climate response itself), improves on this status quo.


Here we propose a compact set of experiments uniquely designed to cleanly assess carbon-climate dynamics relevant for mitigation. Our objectives are threefold:

- Re-assess the transient climate response to cumulative CO2 emissions: assess the response of temperature change and land/ocean carbon dynamics as a function of cumulative emissions which are the independent variable of the
experiment
- Assess the Zero Emissions Commitment across models: systematically measure the unrealized warming that continues after all $CO_2$ emissions have been halted (again, in an experiment where emissions are the independent variable).
- Explore climate reversibility potential: isolate the impacts of global scale carbon removals, assessing hysteresis in
the relationship between climate and cumulative $CO_2$ emissions.

Follow up studies will then consider in more detail the component level and regional responses in the experiments, considering transient responses, commitment and reversibility of ocean heat transport, regional climatology and land carbon dynamics.

## 2. Flat10MIP Experiment design

(Sanderson et al., 2023) proposed 4 new experiments (Figure 1) which would form part of a standard diagnostic suite for
carbon emissions-driven behavior in multi-model comparison activities such as CMIP. These experiments assess behavior under sustained constant carbon emissions, immediate cessation of emissions and climate reversibility under an idealized continuous climate restoration pathway where all emissions are removed by the end of the simulation (Figure 1). In Flat10MIP, we simulate 3 of the 4 experiments proposed in (Sanderson et al., 2023) as a pilot study with CMIP6 generation models in preparation for CMIP7. Below, we briefly describe the experiments as conducted in Flat10MIP, and recommendations for a
protocol in CMIP7 and beyond.



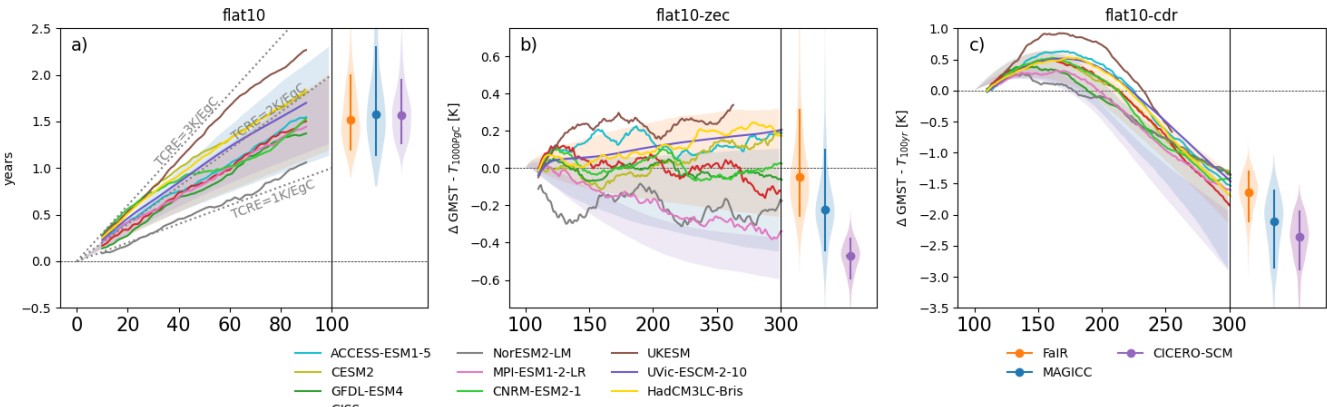

**Figure 1: Experiment design. a) and b) show annual and cumulative carbon emissions as a function of time for the four experiments. Panel c) shows global mean surface temperature derived from cumulative emissions, assuming a perfectly linear TCRE relationship, with expected temperature evolution assuming cumulative emissions proportionality using the IPCC AR6 WGI best TCRE estimate (solid line, 1.65°C per 1000 PgC) and likely range (shaded area, 1.0-2.3°C per 1000 PgC)** (Intergovernmental Panel on Climate Change (IPCC), 2023b).

## 2.1 esm-flat10

The esm-flat10 experiment would serve as an emissions-driven experiment to diagnose the Transient Climate Response to cumulative $CO_2$ Emissions (TCRE), which is the warming from pre-industrial levels observed after the emission of 1000PgC in a transient scenario. The esm-flat10 experiment would branch from a stable esm-piControl simulation, with a constant annual prescribed anthropogenic flux of carbon of 10PgC/year into the atmosphere, with globally homogenous emissions. In esm-flat10, the 1000PgC threshold would occur in year 100 - such that TCRE could be estimated as the time average between global mean warming in years 90-110, sampling over internal variability in this period. 150 years of simulation are requested to allow the simulation to reach 2x pre-industrial $CO_2$ concentrations in most cases (allowing for a wide range of plausible land and ocean carbon uptake). For CMIP7 and future experiments, we recommend a 300 year experiment to allow the simulation to explore potential nonlinearities in response at higher forcing levels.

## 2.2 esm-flat10-zec

The esm-flat10-zec experiment serves as an emissions-driven experiment to diagnose ZEC, which is the additional warming seen a certain number of years after the abrupt cessation of emissions. The esm-flat10-zec experiment would branch from year 100 of the esm-flat10 experiment, with an immediate cessation of emissions and the system is then left to evolve for 220 years. Here, as in (MacDougall et al., 2020) - we use the notation ZEC-n to correspond to the warming *n* years after the cessation of emissions. ZEC50 would thus be the average temperature change relative to that when emissions cease, averaged over a 21 year period, 50 years after the cessation of emissions (i.e. years 140-159). ZEC100 could similarly be calculated using years 190-209). For CMIP7 and beyond, we recommend 300 years for esm-flat10-zec, to allow for longer timescale comparisons with esm-flat10-cdr.





## 2.3 **esm-flat10-cdr**

The esm-flat10-cdr experiment serves as an emissions-driven experiment to diagnose the response of the climate system to
165   reducing, and ultimately reaching net-negative emissions and will provide a measure of climate reversibility when all
cumulative anthropogenic emissions are removed–i.e. reaching the point of cumulative net zero emissions–at the end of the
experiment. The esm-flat10-cdr experiment would branch from year 100 of the esm-flat10 experiment, with a linear ramp
down of emissions (from a starting point of 10PgC/yr) of -0.2PgCyr$^{-1}$ - such that net zero emissions are achieved in year 150
and a negative flux of -10PgCyr$^{-1}$ is achieved in year 200. This negative emission flux of -10PgC/yr would then be held
170   constant from years 200-300, such that by year 300 - cumulative emissions from the start of the simulation would be zero. A
20 year extension follows keeping the emissions at zero.  For CMIP7 and beyond, we recommend 300 years for esm-flat10-
cdr, to allow for better evaluation of system dynamics after the termination of negative emissions.

## 2.4 **esm-flat10-nz**

We propose a final experiment for CMIP7 and beyond, not conducted here *esm-flat10-nz* (Sanderson et al., 2023) which
branches from esm-flat10-cdr in year 150 at the point at which the simulation reaches net zero $CO_2$ emissions, keeping
emissions at zero thereafter.  Such an experiment would provide a proxy for warming commitment after a gradual semi-
idealised emission reduction to net zero, and would provide additional information on ZEC. We recommend that such an
experiment should run ideally for 250 years to allow for comparison of long term dynamics with esm-flat10-zec and esm-
flat10-cdr.


| Experiment | Branches from | Years (this paper) | Years (CMIP7 recommended protocol) | CO₂ emissions | Diagnostic Metrics |
|---|---|---|---|---|---|
| esm-flat10 | esm-piControl | 150 years (From year 0 to year 149) | 300 years | 10PgC/year constant emissions, globally homogenous flux | TCRE |
| esm-flat10-zec | esm-flat10 (branch at start of year 100 of esm-flat10) | 220 years (From year 100 to year 319) | 300 years (From year 100 to year 400) | 0 PgC/yr constant | ZEC50 ZEC100 ZEC200 |



| esm-flat10-cdr | esm-flat10 (branch at start of year 100 of esm-flat10) | 220 years (From year 100 to year 319) | 300 years (From year 100 to year 400) | • Linearly declining emissions by 2PgC/decade from 10PgC/yr (year 100) to -10PgC/Yr (year 200) <br> • Constant -10PgC/yr (years 200-300) <br> • Zero emissions for year 300-320 | TNZ, TR1000 TR0 tPW |
| esm-flat10-nz* | esm-flat10-cdr (branch in year 150) | - | 250 years (From year 150 to year 400) | 0 PgC/yr constant | |

*Table 1: Experiment design for emissions-driven diagnostic runs, detailing the branch point, length and configuration of the experiments as conducted in Flat10MIP (present study). The CMIP7 recommended protocol are run lengths and experiments suggested for future multi-model comparisons, including esm-flat10-nz which not included in flat10MIP.*


The esm-flat10-cdr experiment allows for a number of simple idealized diagnostics which are relevant to the net zero transition and the response of the system to net negative emissions (Fig 2). Like ZEC, each of these metrics is a measure of the path-dependence of the temperature to cumulative emissions relationship, and thus would have a value of exactly zero if global temperature response exactly followed TCRE proportionality. They include

- Temperature difference at net zero (TNZ): This measures the error associated with assuming cumulative emissions proportionality to predict temperatures at net zero. esm-flat10-cdr reaches net zero emissions in year 150, with cumulative emissions of 1250PgC (calculated from year 1, see Figure 1). TNZ is calculated as a 21 year average around year 150 in esm-flat10-cdr (i.e. 50 years after branching from esm-flat10) minus the expected temperature at net zero using cumulative emissions proportionality ($T_{ref}$, which is 1.25 times the esm-flat10 derived TCRE - see
Figure 2).
- Temperature asymmetry under $CO_2$ removal at 1000 PgC (TR1000): This measures the asymmetry in warming during positive and negative emissions at the same net cumulative emissions. It is calculated as a 21 year average around year 200 in esm-flat10-cdr minus a 21 year average around year 100 in esm-flat10. TR1000 would be a measure of hysteresis in global mean temperature when cumulative emissions return to 1000PgC on the downward branch minus
the expectation from TCRE. This could be calculated using a combination of the esm-flat10 and esm-flat10-cdr




experiments for a cumulative carbon emissions total of 1000PgC. esm-flat10-cdr reaches 1000PgC cumulative emissions in year 200 on the downward branch (see Figure 1). esm-flat10 itself reaches 1000PgC in year 100.

- Temperature asymmetry under $CO_2$ removal at 0 PgC (TR0): This is a measure of carbon-climate reversibility when all emitted carbon has been removed from the atmosphere. It is calculated as the average of years 301-320 in esm-flat10-cdr minus mean global temperatures in esm-piControl. TR0 is a measure of hysteresis in global mean temperature when cumulative emissions return to zero after a period of negative emissions. This is calculated using a combination of the esm-piControl and esm-flat10-cdr experiments. esm-flat10-cdr reaches zero cumulative emissions in year 300 on the downward branch (see Figure 1).

- Time to Peak Warming (tPW): This is a measure of the difference in timing between net zero and peak warming. It is calculated as the time difference between the peak value of 20-year smoothed global mean temperatures and the point that net zero is achieved in esm-flat10-cdr (year 150). This metric has a clear policy-relevant translation as the expected time it will take for the climate system to achieve maximum $CO_2$-driven global warming after (or before) reaching net zero emissions under a smooth positive-to-negative emissions transition.

This ensemble provides a broad range of climate model structures and components to evaluate emissions-driven climate reversibility. Each model is selected for its specific configuration, facilitating the exploration of feedback processes and carbon cycle dynamics.

We include 8 CMIP6 generation Earth System Models, one CMIP3 generation model, one intermediate complexity model and the three simple climate model ensembles used in the AR6 IPCC assessment(Forster et al., 2023). The ESMs and SCMs participating in this study are listed in Table 2 and more fully described in the Appendix. Each Earth System Model has completed one ensemble member of each of the MIP experiments (esm-flat10, esm-flat10-cdr and esm-flat10-zec) - with supporting existing experiments from CMIP6 (C4MIP, ZECMIP and CDRMIP). For each SCM, an ensemble of approximately 1000 simulations are completed with simple climate model versions spanning a range of climate responses consistent with assessed climate uncertainty (using a combination of observational constraints, IPCC assessed ranges and ESM data to constrain the parameter space of the simple climate models (IPCC AR6 working group 1: Technical summary, 2023).
 In this study, we summarize the global mean characteristics of the simulations which conducted the experiments, while additional dedicated domain-specific studies will assess regional aspects of transient emissions- driven response and reversibility.







**Figure 2: Schematic of metrics derived from esm-flat10-cdr experiment to quantify different aspects of temperature reversibility under a continuous transition from positive to negative emissions. Dashed lines correspond to temperature trajectories for a hypothetical case where temperatures do not perfectly follow cumulative CO₂ emissions.**



## 3. Results

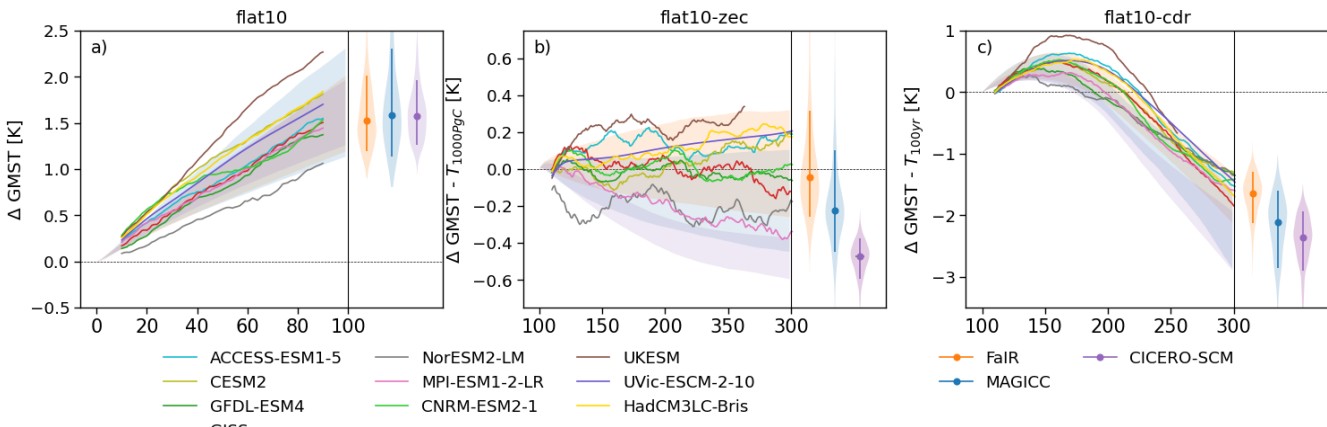

**Figure 3: Summary results for global mean Surface temperature (GMST)response in the trial flat10MIP. Colored lines indicate temperature change from (a) pre-industrial levels (b,c) T100yr (the average temperature in years 91-110 in esm-flat10) in each of the participating ESMs. Shaded regions refer to the Simple Climate Models' probabilistic distribution ranging the 10th–90th percentiles. This distribution is shown as violin plots for the last time step of each scenario, where the shading shows the full range of results, and the vertical line indicates the 10th–90th percentiles with the median in the center. A 20-year moving average is applied to all time series.**

Figure 3 illustrates the global temperature response for the 3 simulations requested in flat10MIP. Throughout this section, we refer by default to $T_{100yr}$ - the warming in units K after 1000PgC of cumulative emissions (which in *esm-flat10* occurs in year 100), numerically equivalent to TCRE which has units K/1000PgC but allowing proper consideration of the arithmetic sum with ZEC-n, also in units K. Summary metrics as defined above for each model, are detailed in Table 2. Figure 3a shows that the range of $T_{100yr}$ seen in the ESM ensembles (1.1K-2.4K) is broadly captured by the SCMs considered in this study, though MAGICC shows a greater upper bound in $T_{100yr}$ (10th-90th percentile of 1.1-2.7K) relative to FaIR or CICERO-SCM (10th-90th percentiles of 1.1-2.1K and 1.2K-2.1K respectively). However, we see differences in the ZEC and reversibility distributions. The ESM ZEC distribution is best captured by FaIR (ZEC100 range of -0.1K to +0.2K), whereas MAGICC and CICERO-SCM simulate more negative values -0.2 to +0.1K and -0.3 to -0.1K respectively. We also see that two of the three SCM ensembles (MAGICC and CICERO-SCM) tend to simulate stronger temperature decline under negative emissions than seen in any of the ESMs, although the FaIR ensemble is broadly consistent. The intermediate complexity model, UVic-ESM lies within the ESM distribution for both $T_{100yr}$ and ZEC100.





## 3.1 Earth System Model responses



**Figure 4: ESM results.** Columns represent global indicators from ESM simulations running esm-flat10 (left), esm-flat10-zec (center) and esm-flat10-cdr (right). Panels a)--c) show changes in GMST with the dashed black line and gray shading denoting central estimate and range derived from cumulative emissions assuming a linear TCRE relationship as given in AR6 (TCRE=1.65 K, likely range 1.0 -- 2.3 K) for reference. Panels d)--f) illustrate changes in atmospheric CO2 concentrations as a function of time. Panels g)-i) show cumulative carbon absorption by the land surface. Panels j-l) show cumulative absorption of carbon by the ocean over time. The circles for the esm-flat10-cdr experiments indicate the maximum of each time series. A 20-year moving average is applied for the GMST time series (bold line), faint line shows original data.

Figure 4 illustrates ESM results in more detail, showing the evolution of a number of climate indicators. In *esm-flat10*, emissions are constant at 10PgC/yr - and thus temperature change from pre-industrial in year 100 is a measure of the Transient Response to Cumulative CO$_2$ Emissions. Figure 4a illustrates the range of transient response in the context of the assessed

TCRE range in IPCC AR6 (1.0 to 2.3 K/(1000PgC)) (Intergovernmental Panel on Climate Change (IPCC), 2023a). The models considered in the present MIP largely span this range, with values of TCRE (as calculated from 1pctCO2) from 1.2K to 2.6K (Table 2). Model land sink evolution varies during the extended esm-flat10 simulations, with some models showing

a saturation of the land sink (HadCM3, UVic, ACCESS), while others show continued land uptake throughout the experiment (CESM, NorESM, GFDL, CNRM).

Figure 4b shows how temperatures evolve in *esm-flat10-zec* - showing that temperatures remain (approximately) stable following cessation of emissions, even though atmospheric carbon dioxide concentrations decline. Different models show a

diversity of evolution of land and ocean carbon sinks - with some models (e.g. MPI, GFDL) continuing to absorb carbon into the land surface during the zero emissions phase, while the majority stabilize rapidly. Similarly, all models indicate a continued uptake of carbon in the ocean during the zero emissions phase.

The reversibility experiment *esm-flat10-cdr* branches from *esm-flat10* in year 100 and linearly reduces emissions by 2PgC /

decade such that net zero occurs in year 150. Linear emissions reduction then continues until year 200 at which point emissions are held at -10PgC/yr until year 300 and set to 0PgC/yr thereafter such that cumulative emissions are zero from year 300 onwards (Fig. 1c). As such, any residual temperature change after year 300 is indicative of non-proportionality of temperature with cumulative emissions. Global mean results are summarized in Figure 4c, showing that peak warming can occur either before or after net zero (but most models peak before), as seen in similar experiments in (Koven et al., 2023). By the end of

the simulation, some models remain warmer than pre-industrial (CESM,CNRM, ACCESS,UVic), while some are cooler (GISS, MPI, NorESM, GFDL).

All models are in agreement that peak $CO_2$ concentrations occur before net zero, and all models predict that the ocean carbon sink peaks after net zero. All models predict that the cumulative ocean carbon sink will decline but stay positive. However,

models disagree on the timing of the land sink relative to net zero. GFDL, CESM, NorESM, GISS, MPI and CNRM show the cumulative land carbon sink peaking after net zero, whereas HadCM3, UVic and ACCESS show the cumulative land carbon sink peaking before net zero. At zero cumulative emissions in year 300, models range from the cumulative land sink being near-zero to being a slight net source of carbon over the 300 year period (model range -50 to 0PgC), while all models agree that the cumulative ocean sink is a net sink (model range 120-220PgC).






**Figure 5:** Evolution of carbon sinks in ESMs in flat10MIP. [Light blue/dark blue/green] shading shows the [airborne fraction/ocean fraction/land fraction] of emissions in each year as a function of time. Gains to each domain are shown in solid colors, while losses are shown in light dotted colors. The left hand column shows the fractions for esm-flat10, where emissions total 10PgC/yr. Central column shows results for esm-flat10-zec, where emissions are zero and atmospheric loss is compensated by gains in the land and ocean. Right hand column shows esm-flat10-cdr, where removals are balanced by losses from each of the pools.



Figure 5 shows how the rate of carbon emission allocation to the atmosphere, land, and ocean evolves as a function of time in the different experiments. In *esm-flat10,* we see a diversity of behavior - although all models rapidly adjust from a high

airborne fraction at the start of the experiment to a greater uptake by land and ocean, the airborne fraction in year 100 varies by model (ranging between 0.45 and 0.55, see Figure 6). We also observe that although most models reach a constant fractional allocation to land, atmosphere and ocean by year 100, there are exceptions - with UKESM and ACCESS showing peak land uptake some decades into the experiment with declining uptake thereafter. During the *esm-flat10-zec* experiment, we see inter model differences in how atmospheric carbon losses are balanced by land or ocean uptake - some models (e.g. GFDL, CNRM)

dominated by land, others (GISS, ACCESS) dominated by ocean. Similarly in *esm-flat10-cdr,* there are large model differences - with some models which show residual warming and some models with net cooling at the end of the experiment.

## 3.2 Global response indicators in flat10 and other experiments

### 3.2.1 Transient response to positive emissions

Figure 6 and Table 2 illustrate the global trajectories and summary indicators of the ESMs which participated in the experiment set in both *esm-flat10* and *1pctCO2* (drawing on results from (Arora et al., 2020)). Figure 6a shows that this compatible emissions timeseries is time-varying and model dependent - with typical behavior showing compatible emissions growing from ~10PgC/yr at the start of the experiment to between 16-22PgC/yr at the time at which cumulative emissions reach 1000PgC. As such, compatible cumulative emissions are weighted towards the end of the experiment - the mean result exceeds

500PgC in year 39 and 1000PgC in year 65 (Figure 6b). Compatible emissions in *1pctCO2* are also significantly greater than current anthropogenic emissions (11.1 $\pm$ 0.8PgC/yr in 2023 (Friedlingstein et al., 2023).





*Figure 6: Comparative ESM results for TCRE calculation using 1pctCO2 and esm-flat10, showing [a,b] compatible*
*[annual,cumulative] emissions in 1pctCO2 compared with the constant 10PgC/yr flux in esm-flat10. Annual total*
*anthropogenic carbon emissions in 2023 are shown for context . [c,d] show temperature evolution in [1pctCO2,esm-*
*flat10]. Colored lines show global model output from available ESMs with a 21 year moving average applied. [e,f] show*
*airborne fraction in [1pctCO2,esm-flat10]. Circles show results at the time when cumulative emissions reach*
*1000PgC. Shaded region in [d] illustrates the range of warming according to the IPCC AR6 assessed likely range of TCRE.*







| Earth System Models | T1000PgC (TCRE from 1pctCO2 at 1000PgC) | T100yr (TCRE from flat10) | ZEC50 (from esm-1pct-brch-1000PgC) | ZEC90 (from 1pct-brch-1000PgC) | ZEC50 (from flat10-zec) | ZEC100 (from flat10-zec) | t-PW | TNZ | TR1000 | TR0 |
|---|---|---|---|---|---|---|---|---|---|---|
| Units | K | K | K | K | K | K | years | K | K | K |
| ACCESS-ESM1-5 (Ziehn et al., 2020) | 1.90 | 1.75 | 0.01 | -0.03 | 0.21 | 0.13 | 7 | 0.08 | 0.23 | 0.17 |
| CESM2 (Danabasoglu et al., 2020) | 2.00 | 1.95 | -0.31 | -0.17 | -0.27 | -0.13 | -10 | 0.05 | 0.03 | 0.42 |
| GFDL-ESM4 (Dunne et al., 2020) | 1.45 | 1.45 | - | - | -0.21 | 0.11 | -29 | -0.09 | -0.25 | -0.11 |
| GISS (Kelley et al., 2020) | 1.6*** | 1.62 | -0.11*** | -0.06*** | 0.19 | -0.24 | -4 | 0.12 | 0.01 | -0.56 |
| NorESM2-LM (Seland et al., 2020a) | 1.32 | 1.18 | -0.33 | -0.32 | -0.23 | -0.31 | -33 | -0.03 | -0.23 | -0.31 |
| MPI-ESM1-2-LR (Gutjahr et al., 2019) | 1.65 | 1.50 | -0.27 | -0.37 | -0.14 | -0.24 | 1 | -0.06 | -0.24 | -0.29 |
| CNRM-ESM2-1 (Séférian et al., 2019) | 1.73*** | 1.72 | 0.06*** | 0.25*** | -0.01 | 0.11 | -10 | 0.11 | 0.03 | 0.38 |
| UKESM1.2 (Sellar et al., 2019) | 2.55*** | 2.45 | 0.28*** | 0.33*** | 0.27 | 0.21 | -3 | 0.2 | 0.48 | - |
| UVic-ESCM-2-10 (Mengis et al., 2020b) | 1.86 | 1.80 | 0.04 | 0.02 | 0.01 | 0.12 | 3.0 | 0.07 | 0.20 | 0.25 |
| HadCM3LC-Bris (Valdes et al., 2017) | 1.93** | 2.02 | - | - | 0.01 | 0.15 | 11.0 | -0.03 | 0.17 | 0.15 |
| **Simple Climate Models** | T100yr (TCRE from 1pctCO2) | T100yr (TCRE from flat10) | ZEC90 (from esm-1pct-brch-1000PgC) | ZEC100 (from esm-1pct-brch-1000PgC) | ZEC50 (from flat10-zec) | ZEC100 (from flat10-zec) | t-PW | TNZ | TR1000 | TR0 |
| MAGICC6 (Meinshausen et al., 2011) | 1.71(1.13,2.68) | 1.59(1.05,2.66) | -0.12(-0.28,0.19) | -0.18(-0.44,0.22) | -0.11(-0.23,0.12) | -0.16(-0.38,0.15) | -6.00(-13.05,8.00) | -0.04(-0.11,0.11) | -0.19(-0.43,0.27) | -0.56(-1.06,0.14) |
| FaIR (Smith et al., 2018) | 1.57(1.16,2.10) | 1.54(1.13,2.18) | -0.02(-0.19,0.34) | -0.04(-0.28,0.48) | -0.02(-0.13,0.25) | -0.03(-0.22,0.37) | 0.00(-11.0,15.0) | 0.01(-0.07,0.17) | -0.03(-0.26,0.48) | -0.15(-0.44,0.36) |
| CICERO-SCM (Sandstad et al., 2024) | 1.69(1.33,2.21) | 1.58(1.21,2.11) | -0.05(-0.13,0.09) | -0.25(-0.33,-0.11) | -0.18(-0.26,-0.09) | -0.34(-0.47,-0.24) | -10.00(-16.0,-4.0) | -0.06(-0.11,0.0) | -0.35(-0.52,-0.20) | -0.79(-1.08,-0.61) |

**Table 2: Summary diagnostics from flat10MIP and ZECMIP/1pctCO2 (MacDougall et al., 2020)** experiments for Earth System Models and Simple Climate Models, which reported $T_{1000PgC}$, ZEC50 and ZEC90. $T_{100yr}$ is the mean warming in years 91–110 in esm-flat10, ZEC50/ZEC100 is the zero emissions commitment measured as the warming in esm-flat10-zec between years 100 and years 150/200 respectively. t-PW is the time difference in years of peak warming in esm-flat10-cdr relative to net zero in year 150, (TNZ/TR1000/TR0) is the warming in years (150/200/310) in esm-flat10-cdr relative to warming in years (125/100/0) in esm-flat10 when cumulative emissions are (1250GtC/1000GtC/0GtC) respectively. *(personal communication, Chris Jones). **(personal communication, Anastasia Romanou). ***1pctCO2 responses are from a similar but non-identical previous model version.







**Figure 7: (a) for ESMs, a comparison of $T_{100yr}$ [flat10] and $T_{1000PgC}$ [1pctCO2], $T_{100yr}$+ZEC50 [flat10-zec]/$T_{1000PgC}$+ZEC50 [1pctCO2] (small transparent points) and $T_{100yr}$+ZEC100 [flat10-zec]/$T_{1000PgC}$+ZEC90 [1pctCO2] (large transparent points) for ESMs participating in flat10MIP (red) and ZECMIP (blue, where available). The final point is the multi-model mean for cases where there exist complete runs for both ZECMIP and flat10MIP [ACCESS, CESM2, NorESM, MPIESM and CNRM-ESM2] (b) for SCMs, violin plots showing distributions of $T_{100yr}$, $T_{100yr}$+ZEC50 and $T_{100yr}$+ZEC100 for esm-flat10 (left) and 1pctCO2 (right).**






Figure 7 compares distributions of $T_{100yr}$ and ZEC computed using the *1pctCO2* and *esm-flat10* approaches. Note that (MacDougall et al., 2020) reported ZEC90 for ESMs contributing to ZECMIP, whereas in Flat10MIP (and in the 1pctCO2

simulations for simple climate models) we report ZEC100. ZEC100 cannot be easily computed from ZECMIP because most models did not complete simulations beyond 100 years after the exceedance of 1000PgC.

We see, on average, a slight offset such that TCRE estimates in the ESMs have a value that is an average of 0.12K greater in 1pctCO2 relative to *esm-flat10* (see Table 1, Figure 7). This is consistent with (Krasting et al., 2014), who found that TCRE estimated at high emissions rates was greater than that estimated using present day emissions rates. Similarly, distributions in

the simple climate models MAGICC and CICERO-SCM, show $T_{1000PgC}$ from 1pctCO2 is on average about 0.1K greater than $T_{100yr}$ from *esm-flat10*. The third simple climate model, FaIR, shows comparable values of $T_{1000PgC}$ and $T_{100yr}$ (Figure 7). Figure 8a shows correlations between $T_{1000PgC}$ and $T_{100yr}$ - re-enforcing the small average offset between the two approaches - though the gradient of the best fit line is near-unity.

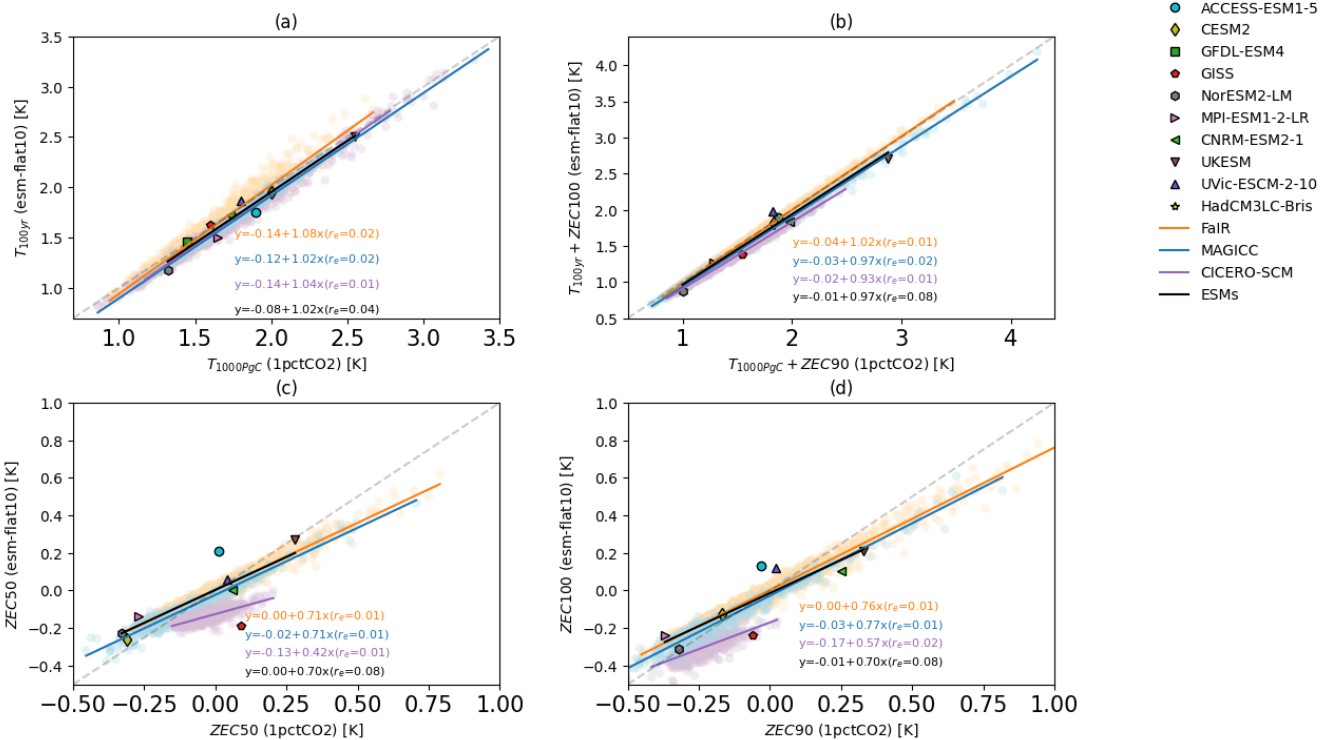


**Figure 8: Comparison metrics assessed using the flat10MIP methodology and 1pctCO2 based experiments. ESM summary metrics are $T_{100yr}$, ZEC50 and ZEC100 for esm-flat10 and $T_{1000PgC}$, ZEC50 and ZEC90 for 1pctCO2. Filled shapes illustrate values assessed from ESMs, pale dots illustrate members of the Simple Climate Model ensembles for (FaIR, MAGICC, CICERO-SCM) in (orange,**

**blue, purple). Straight lines show least-square best fits for the ESMs (black) and SCMs.**



### 3.2.2 Zero emissions commitment

For ZEC, however, we see greater differences between the concentration-driven approach and the emissions driven approach

than for TCRE (Figures 7,8,9). In the SCM ensembles, ZEC50 and ZEC100 are of order 25% smaller if measured using the flat10-zec protocol relative to the ZECMIP protocol (this is true irrespective of whether ZEC is positive or negative, Figure 8c,d). We also note that one SCM, CICERO-SCM, shows more consistently negative values of both ZEC50 and ZEC100 when quantified via flat10MIP than by ZECMIP (Figure 8c,d). ESMs are also consistent with the relationship of ~25% smaller absolute magnitudes in ZEC50 and ZEC100, albeit with larger scatter. Some models (NorESM, CESM2, MPI, CNRM) in the

ZECMIP experiment suggest an apparent short term warming pulse, followed by a cooling in the decades following cessation, which is less apparent in the *esm-flat10-zec* experiment (Figure 9) - but additional ensemble members are required to properly quantify this behavior. In the MPI model, this is consistent with findings that TCRE was higher using the ZECMIP protocol compared to flat10MIP (Fig. 1d in (Winkler et al., 2024)).

It is also evident that *total* warming measured from pre-industrial levels 100 years after emissions cease (i.e. $T_{100yr}$+ZEC100 from *esm-flat10* and $T_{1000PgC}$+ZEC100 from *1pctCO2*), are more consistent between ZECMIP and flat10MIP protocols (Figure 8b) than either of TCRE or ZEC100 independently - indicating that total warming following a period of emissions followed by cessation is path independent in the models considered here. However, we continue to see in the mean values of the SCM distributions (Figure 7b) for MAGICC and CICERO-SCM that $T_{1000PgC}$+ZEC100 is ~0.1K greater in *esm-flat10-zec* than in

*esm-1pct-brch-1000PgC*. FaIR is again consistent between the two approaches, with only 0.01K difference between mean values. For the ESMs (Figure 7a), we note that multi-model mean $T_{1000PgC}$+ZEC100 is 0.05K greater for *esm-1pct-brch-1000PgC* than $T_{100yr}$+ZEC100 for *esm-flat10-zec* (wheras mean $T_{1000PgC}$ is 0.12K greater than $T_{100yr}$).

Our results in general suggest that the weighting of compatible emissions towards the end of the simulation in *1pctCO2*, as

well as the shorter total time period over which emissions occur in *1pctCO2* (~70 vs 100 years), have an impact on both the estimate of TCRE and the transient response following cessation of emissions. We tend to see slightly greater estimated values of TCRE in *1pctCO2,* with most models exhibiting short term continued warming, followed by cooling in the decades following cessation of emissions. In contrast, behavior in *esm-flat10-zec* has slightly less warming during the positive emissions phase, and less adjustment afterwards, resulting in lower values for TCRE and smaller magnitudes (either positive

or negative) of ZEC.

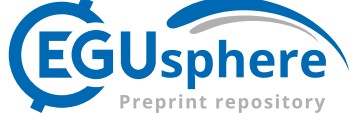



**Figure 9: global mean temperature change evolution for ESMs participating in flat10MIP (bold colors), in the context of 1pctCO2 (grey) and ZECMIP where comparable simulations with the same model version are available (faded colors) . Red lines show the positive emissions period (10PgC/yr for flat10, solid red and 1pctCO2 compatible emissions for ZECMIP), blue/grey lines show zero emissions period for esm-flat10-zec and esm-1pct-brch-1000PgC respectively. Horizontal dashed lines show [T$_{100yr}$,T$_{1000PgC}$] as estimated from [esm-flat10 (red),1pctCO2 (grey)].**



3.2.3    **Climate Reversibility Experiments**

4.

**Figure 10: Global mean temperature relationship with cumulative emissions for the ESMs.** A 21-year moving average is applied for the GMST time series. Arrows show the direction of time, with [red,yellow,blue] lines showing [constant positive, rampdown, constant negative] phases of the experiment. Black dashed and dotted lines show TCRE * cumulative emissions ($I_{em}$) and TCRE * 405    cumulative emissions + ZEC100 for each model, using TCRE and ZEC100 values as calculated from esm-flat10 and esm-flat10-zec.



Figure 10 illustrates global scale hysteresis in the ESM results, showing the change in global mean surface temperature as a function of cumulative emissions. Though all models broadly indicate proportionality between temperature and cumulative emissions, there are some notable deviations. Many models indicate some hysteresis, either positive (ACCESS) or negative
(GFDL, NorESM, MPI-ESM), between the upward and downward branches of the simulation, and some (CESM2, GFDL, CNRM) appear to show a change in temperature/cumulative emissions response during the course of the downward branch. Overlain in dotted lines on each panel of figure 10 is a null hypothesis, informed only by TCRE and ZEC100 from the *esm-flat10* and *esm-flat10-zec* experiments, that temperatures in the net-negative emissions period of *esm-flat10-cdr* might be explained as a combination of the TCRE * $I_{em}$ + ZEC terms (Koven et al., 2022, 2023). This framework explains much, but
not all, of the hysteresis observed; in particular some of the models (e.g. GFDL, UKESM) show larger hysteresis than predicted by ZEC100, and the TCRE+ZEC framework does not predict the deviations late in the downward branch for those models which have such dynamics.

Figure 11 shows how additional climate indicators vary with cumulative emissions. Atmospheric carbon dioxide levels are
consistently lower on the downward branch, but cumulative land carbon sink hysteresis varies by model - with some models showing significantly larger cumulative land carbon sinks on the downward branch (e.g NorESM), while some models (e.g. GISS, HadCM3LC) show cumulative sinks proportional to cumulative emissions on both upward and downward branches. Similarly, all models show hysteresis in cumulative ocean sink strength with cumulative emissions, with between 100 and 200PgC remaining in the ocean in year 300 of *esm-flat10-cdr*.


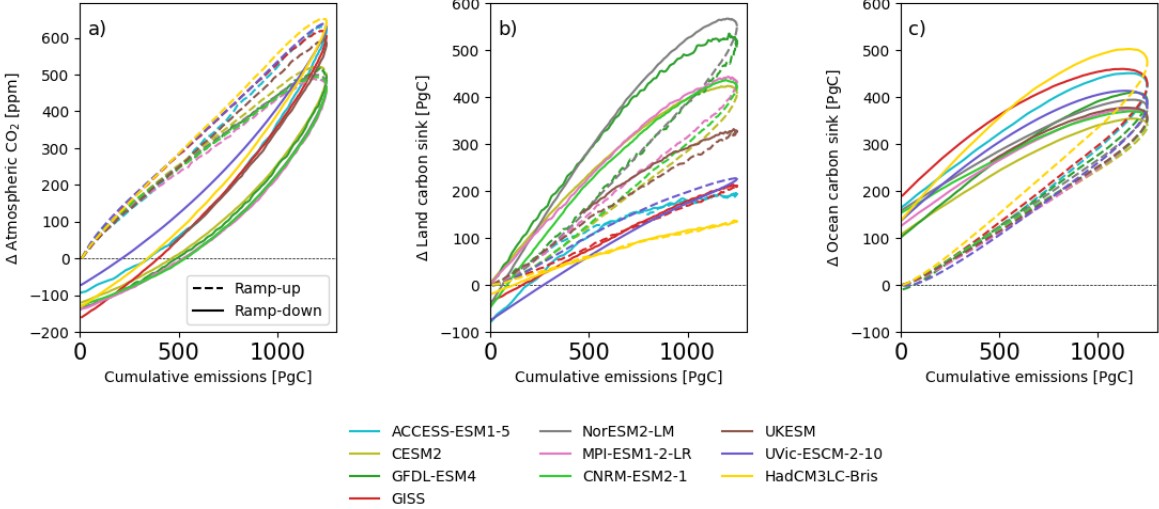

**Figure 11: Climate indicators as a function of cumulative emissions for the ESMs. A 21-year moving average is applied for the all time series.**



We identify a number of new metrics (TNZ, TR1000, TR0, and tPW; Fig. 2, Table 2), which are aimed to capture aspects of climate reversibility and commitment from the *flat10-cdr* experiment. As noted above, each of these measures a distinct aspect of potential deviation from perfect TCRE proportionality, and thus, like ZEC, would have a value of exactly zero if temperature were exactly proportional to cumulative emissions.


**Figure 12. Matrix of relationships between metrics quantified here. Shown are pairwise plots between the following metrics: TCRE (T100yr for flat10 and T1000PgC for 1pctCO2), ZEC50, ZEC100 (ZEC90 for ESM results), tPW, TNZ, TR1000, TR0. SCM ensembles are shown as contours at the 10th percentile of the joint distribution for each pairwise comparison (such that 90% of points lie within the contours, FaIR, MAGICC, CICERO-SCM in orange, green, blue, respectively), ESMs are shown as individual**
**points. Diagonal panels show histograms (SCMs) and discrete values (ESMs) for each of the metrics diagnosed here.**



Two of these metrics measure the hysteresis around the net zero transition: tPW is the time offset of peak warming relative to net zero, whereas TNZ is the difference in realized temperature at net zero relative to what one would predict through TCRE proportionality. Fig. 12 shows how these metrics relate to each other, and to TCRE and ZEC. With the exception of TCRE, all metrics show a positive correlation with all other metrics, particularly for SCMs. ESMs show greater scatter across a number of the pairwise relationships than the SCMs, reflecting a greater diversity of potential dynamics arising from their high complexity than are being captured in the more parsimonious relationships represented by the SCMs . For example, in each of the SCM ensembles tPW and TNZ are highly and consistently related - but a number of ESMs (CNRM, GISS, CESM, UKESM) lie outside of the SCM distributions, such that we see peak warming significantly before net zero with greater warming than one would expect from cumulative emissions proportionality (contours in Figure 12 indicate 90% of the ensemble distribution, for tPW vs TNZ, ESM results lie outside of the 99th percentile, not shown). Similar differences are seen in the relationship between tPW and ZEC50. This hints at behavior in the ESMs which might not be represented in current generation of SCM parameter ensembles. This could potentially be due to a number of different processes, e.g. ocean circulation processes such as AMOC weakening which are not represented in current SCMs (Schwinger et al., 2022) but larger ESM initial condition ensembles are necessary to have confidence in the ESM metrics in the presence of internal variability. This discrepancy could potentially be related to studies which have found inconsistencies between the temporal dynamics of the ocean heat and carbon uptake in ESM and SCM ensembles (Séférian et al., 2024), and would benefit from further investigation.

Another pattern that emerges in Fig. 11 is the greater correlation captured in the short-term metrics (ZEC50, TR1000) than in the longest-term metric (TR0) that shows greater scatter with the ZEC and other reversibility metrics. This high correlation (e.g., between ZEC50 and TR1000 and between ZEC100 and TR1000) has an important implication: that most of the uncertainty present in the reversibility of GMST (although not necessarily regionally or in other metrics (Schleussner et al., 2024)) under an idealized overshoot scenario will also be present under zero emissions at the same level of cumulative emissions that avoids the overshoot.

## 4. Summary and Conclusions

The finding of a near-linear relationship between cumulative carbon emissions and global mean temperature (Allen et al., 2009; Matthews et al., 2009) enabled recent climate policy to link desired limits for warming to an allowable budget of remaining carbon emissions. The years following have seen regular efforts to quantify remaining carbon budgets for the Paris Agreement goals (Lamboll et al., 2023), with scenarios built on this premise (Rogelj et al., 2019a), and refinement in the treatment of how to incorporate non-CO2 emissions into this framework (Cain et al., 2019; Jenkins et al., 2018; Mengis and Matthews, 2020).





Further, an increased understanding has emerged that the TCRE relationship is an approximation, owing to fortuitous
cancellation of terms in heat and carbon uptake in many models, but that this cancellation is not perfect and a "Zero Emissions
Commitment" or ZEC (Palazzo Corner et al., 2023) may result in residual carbon-induced warming (or cooling) even if carbon
emissions are held at net zero. This ZEC effect may cause peak temperatures to be seen before or after net zero (Koven et al.,
2023). Building confidence in this timing is important; if peak temperatures occur after net zero, this may create climate
adaptation challenges which might not otherwise be planned for if simple TCRE proportionality is used to predict warming
outcomes. Such non-TCRE dynamics are also related to the emissions levels compatible with a stable climate - which could
potentially be net positive or negative (Jenkins et al., 2022).

Operational methods of quantifying TCRE and ZEC to date have utilized existing default Earth System Model diagnostic
experiments which have focussed on the response of the Earth System to a prescribed concentration pathway - generally an
exponential increase of 1 percent per year - as an idealised proxy for climate change induced by carbon dioxide. It is then
possible to calculate compatible $CO_2$ emissions, specific to a given model, to frame the output of these experiments in terms
of emissions (Jones et al., 2016; Liddicoat et al., 2021) and calculate TCRE, with branched zero-emission experiments to
calculate ZEC (Jones et al., 2019).

Although these experiments have been highly useful in helping to quantify TCRE and ZEC efficiently using mostly pre-
existing simulations, the use of a concentration-driven diagnostic runs has limitations (Gregory et al., 2015; MacDougall,
2019) - emissions are specific to a given model, and are highly weighted towards the end of the experiment when emissions
rates greatly exceed present day or projected levels. As such, given that experiments to measure ZEC seek fundamentally to
measure subtle, second order effects - there is an argument for new diagnostic experiments which cleanly measure TCRE,
ZEC and climate reversibility using reproducible and cleanly interpretable benchmarks.

In this study, we have demonstrated the utility of a new set of idealized experiments that can be applied with both complex
and simple Earth System Models. This 'flat10' framework is based upon a small number of variants around a simple core
experiment, where emissions are fixed at 10PgC/yr for 100 years - a rate which approximates current anthropogenic carbon
emissions, and conveniently totals 1000PgC after 100 years of simulation, with the temperature in year 100 thus providing a
direct assessment of TCRE. Branch experiments from this point can measure the Zero Emissions Commitment (with emissions
set to zero in year 100), and climate reversibility (with an idealized net zero and net negative emissions pathway in which
cumulative emissions reach zero by the end of the experiment). Along with these experiments, we propose diagnostic
measures which serve to measure different aspects of non-TCRE behavior and how they relate to the likely outcomes of real
world net zero and net negative emissions proposals. These experiments complement similar experimental design being
developed and run by the Tipping Point intercomparison project, TIPMIP (Colin Jones et al., in prep). TIPMIP experiments
also follow a prescribed constant $CO_2$ emission pathway, but the emissions are tailored for each model to result in a common

warming rate of 2°C per century. As such, the goal of TIPMIP is to examine the behavior of ESMs at common levels of global warming, while the goal of the flat10MIP experiments is to examine the behavior of ESMs under common external

forcing. Furthermore, for future experiments using the TIPMIP protocol, the flat emissions pathway in esm-flat10 will likely provide a more accurate TCRE estimate for calibrating the emissions rate required for constant warming rates.

These experiments form part of the 'fast track' recommendation for CMIP7, through which the climate change research community will gain a greater understanding of ZEC and reversibility behavior in the next generation of climate models.

Here, to illustrate the potential for these simulations to diagnose a broad suite of climate response metrics, we demonstrate the results of the flat10MIP experiments for a subset of CMIP6-generation models and the simple climate models used in the IPCC 6th Assessment Report. We find, as expected, that TCRE is first order consistent whether calculated using the 1pctCO2 simulations, or using esm-flat10 simulations - but also that the values of ZEC estimated with 1pctCO2 tend to be greater than for esm-flat10-zec, indicating that the weighting of emissions towards the latter part of the 1pctCO2 experiment may increase

transient warming or cooling trends, potentially driving a larger ZEC than would be seen in a realistic emissions scenario.

We also find a large diversity of ESM behavior in the climate reversibility experiment *esm-flat10-cdr,* including that peak warming can occur before or after net zero emissions and is not necessarily predictable from a combination of TCRE and ZEC (consistent with existing studies(Asaadi et al., 2024)) with a range of carbon sink evolutions in different ESMs, both in the

positive and negative emissions phases of the experiment. Models strongly disagree on the timing and amplitude of peak land carbon uptake, some showing peak uptake decades before and others decades after the net zero transition. There is also evidence of state-changes during the negative emissions phase, with some models showing a change in the rate of cooling per unit carbon removed - potentially indicating dynamical changes in ocean circulation which might impact carbon-climate dynamics.


However, in this study our scope for understanding this diversity is limited: we present the experimental design for CMIP7 plus global scale results from ESMs and SCMs which are now available to the community. Detailed process understanding will be presented in follow-up studies, considering land and ocean dynamical processes from the flat10MIP ensemble, where we hope for wide community engagement.


We argue that emissions-driven diagnostic experiments are the cleanest method for diagnosing the response to climate forcers on a range of relevant timescales. In future, we would imagine these experiments becoming elements of a wider set of idealized, yet policy relevant emission-driven experiments which can efficiently categorize either a simple or complex climate model's response to climate forcers.




In the present study, this has been limited to a specific trajectory of carbon emissions which has been chosen pragmatically to minimize computational burden. Future understanding would be increased by adding to this archive, both in terms of larger initial condition ensembles to improve confidence in ZEC and reversibility metrics, perturbed parameter ensembles in ESMs to understand conditionalities on model calibration choices, and with longer simulations to understand longer timescales of commitment.

Despite these caveats, the present effort has indicated that some models exhibit non-linear and threshold behavior. Further experiments would be required to fully document the conditions under which such transitions occur. As such, future CMIP activities might consider a range of flat-n type experiments spanning warming levels and decarbonisation rates to categorise the response of the carbon-climate dynamics to different types of overshoot pathway. Also, as ESMs increasingly seek to represent the response to a range of activities (land use change, methane, nitrous oxide emissions amongst others), it will become necessary to cleanly categorize the response to each of these in a reproducible fashion - creating a necessity for well-crafted experiments to cleanly represent model responses to non-fossil-$CO_2$ forcers. A shift towards emissions-driven modeling is essential to produce relevant climate simulations for increasingly specific emissions pathways referred to in climate policy, and this requires a new generation of emissions-driven diagnostic experiments.




# Appendix

**Participating Models**

**Earth System Models**

The flat10MIP experiments are included in the recommended CMIP7 'fast track'- a subset of experiments highlighted for particular relevance as input for climate change assessments. In preparation for this recommendation, a trial model intercomparison conducted the esm-*flat10* experiment set for a collection of eight Earth System Models from the CMIP6 ensemble (Eyring et al., 2016), and one Intermediate Complexity Model.

**ACCESS-ESM1-5** (Ziehn et al., 2020)**:**

**Atmosphere:** UM7.3 (Walters et al., 2019)at 1.875° × 1.25° resolution

**Ocean:** MOM5 (Griffies, 2012) at 1° × 1° resolution

**Land:** CABLE2.4  (Kowalczyk et al., 2013)

ACCESS-ESM1-5 features a coupled carbon-nitrogen-phosphorus cycle in the land component (CABLE2.4), with an ocean provided by the GFDL MOM5 model.

**CESM2** (Danabasoglu et al., 2020)**:**

Atmosphere: CAM6 (Bogenschutz et al., 2018) at 1° resolution

Ocean: POP2 (Smith et al., 2010) at 1° × 1° resolution

Land: CLM5 (Lawrence et al., 2019)

CESM2 includes updated aerosol-cloud interactions in CAM6, while CLM5 provides new parameterizations for carbon and nitrogen interactions in terrestrial ecosystems, and POP2 emphasizes ocean-ice dynamics.

**GFDL-ESM4** (Dunne et al., 2020)**:**

Atmosphere: AM4.1 (Horowitz et al., 2020)at 1° × 1° resolution

Ocean: MOM6 (Adcroft et al., 2019)

Land: LM4.1 (Shevliakova et al., 2024)



GFDL-ESM4 uses MOM6 for advanced representations of ocean circulation and biogeochemical processes, with AM4.1 providing a fully coupled aerosol and cloud interaction system. LM4.1 emphasizes nutrient constraints on land carbon cycles.

**GISS-E2-1-G** (Kelley et al., 2020):

Atmosphere: ModelE (Schmidt et al., 2014) at 2° × 2.5° resolution (Kelley et al 2020)

Ocean: GISS Ocean v1 at 1° × 1° resolution

Land: The vegetation model is the Ent Terrestrial Biosphere Model (Kiang et al 2012) with prescribed leaf area index and prescribed interannual variation of land use and land cover (LULC) change; interactive with carbon cycle (Ito et al, 2020)

Ocean carbon: NASA Ocean Biogeochemical Model (GISS version NOBMg, Romanou et al 2013; Ito et al, 2020; Lerner et al 2021)


**HadCM3LC-Bris**

Atmosphere: HadAM3 (Pope et al., 2000), 3.75° x 2.5° resolution, 19 vertical levels

Ocean: HadCM3L (Cox et al., 2000) , 3.75° x 2.5° resolution, 20 vertical levels

Land: MOSES-2 (Essery et al., 2003), with dynamic vegetation  and 9 plant functional types (Cox, 2001)

Ocean BGC: HadOCC(Palmer and Totterdell, 2001) marine biogeochemistry with NPZD biology model.

HadCM3LC-Bris is based on the HadCM3 climate model (Gordon et al., 2000) adapted for use with an interactive carbon cycle by adopting lower ocean resolution (Cox et al., 2000), and subsequently modified slightly for use on Bristol HPC (Valdes et al., 2017).

**NorESM2-LM** (Seland et al., 2020)

Atmosphere: CAM6  (Bogenschutz et al., 2018)  at 2° × 2° resolution (with modifications)

Ocean: BLOM-iHAMOCC (Tjiputra et al., 2020)

Land: CLM5 (Lawrence et al., 2019)

NorESM2-LM shares land and some atmosphere elements with CESM2, but modifies CAM6 to include updated aerosol and

cloud microphysical schemes and uses the isopycnal-coordinate BLOM for ocean processes, which improves deep ocean mixing simulations.

**MPI-ESM1-2-LR**(Mauritsen et al., 2019; MPI, 2024):

Atmosphere: ECHAM6.3 at 1.875° × 1.875° resolution

Ocean: MPIOM (Jungclaus et al., 2013)at 1.5° × 1.5° resolution



Land: JSBACH3 (Reick et al., 2021)

MPI-ESM1-2-LR utilizes ECHAM6.3, featuring updates in atmospheric chemistry processes, while MPIOM improves ocean heat transport. JSBACH3 integrates biogeophysical and biogeochemical interactions.

**CNRM-ESM2-1** (Séférian et al., 2019)**:**

Atmosphere: ARPEGE-Climat version 6 (Roehrig et al., 2020) at 1.4° × 1.4° resolution

Ocean: NEMO (Madec et al., 2017) version 3.6 at 1° × 1° resolution

Land: ISBA (Decharme et al., 2019)

CNRM-ESM2-1 features NEMO 3.6, which includes advanced parameterizations of ocean mixing, and ARPEGE-Climat for atmospheric dynamics, with updates in stratospheric processes and land-atmosphere coupling through ISBA.

**UKESM1** (Sellar et al., 2019)**:**

Atmosphere: HadGEM3-GA7.1 (Walters et al., 2019) at 1.875° × 1.25° resolution

Ocean: NEMO3.6 (Madec et al., 2017) at 1° × 1° resolution

Land: JULES (Best et al., 2011)

UKESM1 includes JULES, which features dynamic vegetation and coupled nitrogen cycles, along with HadGEM3-GA7.1 which provided improved stratosphere-troposphere interactions and cloud-aerosol physics relative to previous versions

**Intermediate Complexity Models**

UVic ESCM 2.10 ((Mengis et al., 2020)):

Atmosphere: 2D energy moisture balance model 3.6° x 1.8° (Fanning and Weaver, 1996)

Ocean: MOM2 3.6° x 1.8° (Pacanowski, 1995) with thermodynamic-dynamic sea ice model (Bitz et al., 2001)

Land: Dynamic vegetation with 5 plant functional types (Meissner et al., 2003); 14 layers of soil; permafrost (MacDougall and Knutti, 2016); no N, P cycle

Ocean: NZPD model with 2 nutrients (N, P) and Fe limitation scheme (Keller et al. 2012)

**Simple Climate Models**

We also include simulations from three Simple Climate Models which provided climate assessments in the IPCC AR6 WG3 assessment (Intergovernmental Panel on Climate Change (IPCC), 2023c).



**MAGICC6** (Meinshausen et al., 2011)**:**

MAGICC6 is a reduced-complexity model that uses simplified representations of global carbon cycles and radiative forcing, allowing for rapid simulation of emissions-driven climate pathways.

**FaIR** (Smith et al., 2018)**:**

FaIR uses simplified equations to model temperature responses and radiative forcing - using pulse-response assumptions to model carbon and thermal responses to climate forcers, with flexible configurations that allow it to mimic the behavior of more complex models in emissions-driven scenarios.

**CICERO-SCM** (Sandstad et al., 2024)**:**

CICERO-SCM is a reduced-complexity model that focuses on simplified representations of carbon cycle and climate feedbacks, but with extensively developed short lived climate forcer parameterisations. It emphasizes flexibility in handling uncertainties in emissions scenarios and climate sensitivity. Calibration and run-scripts for Flat10MIP are archived here (Sanderson and Sandstad, 2024)

**Code availability**

All code to reproduce plots in this study is permanently available at:

10.5281/zenodo.14012042

**Data availability**

All data to reproduce this study is included at:

10.5281/zenodo.14012042

**Author contribution**: Analysis/plots were performed by BMS, NS, CK. Model simulations were conducted by TI, CDJ, TK, HL, PL, SL, NM, ZM, AR, MS, JS, RS, LS, CS, JT, BMS and TZ. Framing and scoping was performed by BMS, VB, TI, CDK, DML, AM, EOR, IRS, ALSS.

**Competing interests**:

Some authors are members of the editorial board of journal GMD

**Acknowledgements**

BMS, RS and ZN acknowledge support from the European Union's Horizon 2020 research and innovation programme under Grant Agreement N° 101003536 (ESM2025). BMS and MS acknowledge the Research Council of Norway under grant agreement 334811 (TRIFECTA). BMS and NS acknowledge support from the European Union's Horizon 2020 research and



innovation programme under Grant Agreement 101003687 (PROVIDE). CDK acknowledges support by the Director, Office of Science, Office of Biological and Environmental Research of the US Department of Energy under contract DE-AC02-05CH11231 through the Regional and Global Model Analysis Program (RUBISCO SFA). ALSS acknowledges support from the National Science Foundation under grant number AGS-2330096 and the US Department of Energy Regional and Global

Model Analysis Program under grant number DE-SC0021209. The work of DML, PL, and IRS is supported by the NSF National Center for Atmospheric Research, which is a major facility sponsored by the NSF under Cooperative Agreement No. 1852977. DH and NM are grateful to be funded under the Emmy Noether scheme by the German Research Foundation (DFG) in the project 'FOOTPRINTS - From carbOn remOval To achieving the PaRIs agreemeNt's goal: Temperature Stabilisation' (project number 459765257). AR acknowledges support from NASA-Modeling Analysis and Prediction (NASA-MAP)

program under grant NNX16AC93_G. AHMD is supported by the Natural Science and Engineering Research Council of Canada Discovery grant program. TI and HL acknowledge support from the European Union's Horizon 2020 research and innovation program (4C, grant no. 821003; ESM2025, grant no. 101003536) and the Deutsche Forschungsgemeinschaft (Germany's Excellence Strategy – EXC 2037 "CLICCS – Climate, Climatic Change, and Society" – project no. 390683824). The MPI-ESM1-2-LR simulations used resources of the Deutsches Klimarechenzentrum (DKRZ) granted by its Scientific

Steering Committee (WLA) under project ID bm1124. RS acknowledges support from the European Union's Horizon Europe research and innovation programme under grant agreement No 101081193 (OptimESM). TZ receives funding from the Australian Government under the National Environmental Science Program (NESP).

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
