# Peer review of "flat10MIP: An emissions-driven experiment to diagnose the climate response to positive, zero, and negative CO2 emissions"

_EGUsphere, 2024_

## Referee Report (RR1)

**Review round 2 – egusphere-2024-3356**
12/05/2025

I thank the authors for their responses to the previous round of reviewer comments. I am glad to see the revisions made to the text which address both my and other reviewers' concerns well.

I have a single additional point I would like to see revised before the paper is accepted. In the abstract the claim is made that "Further, we find Simple Climate Models (SCMs) tend to over-estimate temperature reversibility compared with ESMs". I completely agree that SCMs are shown to overestimate the extent of reversibility in comparison to more complex models of the earth system under flat10MIP experiments.

However, I do not agree the results in this study warrants a blanket statement on the ability of SCMs to capture reversibility following net zero. In the main text you offer a more nuanced discussion of this result in places, including noting that it is unclear how much of your result is a consequence of the parameter ensemble chosen for each SCM, as opposed to a consequence of the structure of SCMs being incapable of capturing hysteresis in reversibility experiments. It would be good to adapt the statement in the abstract, and have a quick check in the rest of the text, to make it clear that you are describing the ability of the standard, or historically-constrained, parameter distributions in SCMs to capture the reversibility characteristics of ESMs, and not necessarily a comment on the ability of SCMs to capture these ESM behaviours overall.

I am happy to accept the manuscript with these revisions.

---

## Author Response (AR2)

Response to review round 2.

Many thanks to the reviewer for the positive assessment.

*I have a single additional point I would like to see revised before the paper is accepted. In the abstract the claim is made that "Further, we find Simple Climate Models (SCMs) tend to over-estimate temperature reversibility compared with ESMs". I completely agree that SCMs are shown to overestimate the extent of reversibility in comparison to more complex models of the earth system under flat10MIP experiments.*
*However, I do not agree the results in this study warrants a blanket statement on the ability of SCMs to capture reversibility following net zero. In the main text you oPer a more nuanced discussion of this result in places, including noting that it is unclear how much of your result is a consequence of the parameter ensemble chosen for each SCM, as opposed to a consequence of the structure of SCMs being incapable of capturing hysteresis in reversibility experiments. It would be good to adapt the statement in the abstract, and have a quick check in the rest of the text, to make it clear that you are describing the ability of the standard, or historically-constrained, parameter distributions in SCMs to capture the reversibility characteristics of ESMs, and not necessarily a comment on the ability of SCMs to capture these ESM behaviours overall.*

Thanks for this point. We agree that the previous version perhaps implied that the bias was inherent to SCMs, rather than potentially a result of an absence of calibration targets relevant to ZEC and reversibility in operation pipelines used for SCMs. We've revised the sentence in the abstract as follows:

"Further, we find existing probabilistic Simple Climate Model(SCM) ensembles tend to over-estimate temperature reversibility compared with ESMs, highlighting the need for additional constraints. "

With similar refinements elsewhere in the document.

---

## Author Response (AR3)

**Review response to egusphere-2024-3356**

Many thanks to both reviewers for the detailed assessment and feedback of our paper.

**Summary of Changes**

- Improved clarity and precision in writing: Numerous edits throughout the manuscript to correct typos, clarify language, and streamline sentence structure based on reviewer suggestions.
- Enhanced process-based interpretation: Expanded discussion of the physical mechanisms underlying observed differences in TCRE, ZEC, and reversibility metrics between 1pctCO2 and esm-flat10 experiments.
- Clarification and qualification of ZEC metrics: All references to Zero Emissions Commitment (ZEC) now specify the timescale (e.g., ZEC50, ZEC90, ZEC100), and comparisons with 1pctCO2-based ZEC metrics are more rigorously framed.
- Expanded discussion of RAZE: A consolidated section now defines the Rate of Adjustment to Zero Emissions (RAZE), relates it to ZEC, and contextualizes it using recent literature.
- More rigorous discussion of reversibility and lag effects: Text now acknowledges the limitations of using flat10-cdr to isolate the effect of negative emissions and includes discussion of confounding effects from earlier emissions.
- Clarified experiment design and rationale: Greater detail on the purpose of each experiment in flat10MIP, including justification for future experiments such as esm-flat10-nz.
- Revised and clarified figure captions and axis labels: Addressed reviewer feedback on inconsistencies, missing units, and visual presentation (e.g., Figures 2, 3, 6, 10, 12).
- Quantitative additions to figures and tables: Added metrics such as ZEC90 to Table 2 and figures; updated figure panels to include all relevant diagnostics.
- Addressed reviewer-suggested literature: Incorporated new references (e.g., Gillett et al., 2013; MacDougall et al., 2020; Jenkins et al., 2022) and cited them where appropriate.
- Reframed policy-relevance of results: Clarified implications of metrics like TNZ, TR0, TR1000, and tPW for understanding warming dynamics relative to net-zero targets.
- Restructured introduction and methods: Reorganized for greater logical flow, including clearer presentation of the motivation, goals, and experimental design of flat10MIP.

**flat10MIP: Response to Reviewer 1**

*The study tests the utility of a set of idealized experiments – the suite of emissions driven "flat10" experiments described in Sanderson et al., 2024 – using an ensemble of ESM, model of intermediate complexity and simple climate model simulations. A set of metrics (TCRE, ZEC and metrics for reversibility) is evaluated and compared to metrics evaluated using a prior set of simulations based on the concentration-driven 1pctCO2 experiment. The study finds that TCRE is slightly lower and the ZEC substantially lower than when quantified with the 1pctCO2 experiment.*

*This is an insightful study which demonstrates that the flat10 experimental protocol is suitable for quantifying policy-relevant metrics in a consistent way, and that these metrics exhibit important differences relative to those quantified from simulations based on the 1pctCO2 experiment. The manuscript is well structured and well written for the most part, but lacks clarity and precision in some instances, which are identified in the specific comments below. The manuscript would also benefit from a stronger processed-based interpretation of the differences relative to metrics quantified from the 1pctCO2 experiment. There are several typos, missing definitions and omissions in the figure labelling (only partly reported below) and the authors should re-read the manuscript carefully before resubmission.*

Many thanks for the positive assessment.  We have endeavoured to improve the process-based interpretation of metrics, and have carefully proofread the manuscript.

**Specific comments**

*43: "...CMIP DECK 1 pctCO2 experiment, where emissions are strongly weighted towards the end of the experiment". This wording is used throughout the manuscript but its meaning is unclear to me. Why are emissions "weighted"? Also, metrics are typically quantified around year 70 of the 1pctCO2 experiment, not at the end.*

We agree that the wording is confusing.  Modified as follows:

The emissions-driven experiments provide consistent independent variables simplifying simulation, analysis and interpretation with emissions rates more comparable to recent levels than existing protocols using model-specific compatible emissions from the CMIP DECK *1pctCO2* experiment, where **emissions rates tend to increase during the experiment, such that at time of CO2 doubling in year 70, emissions are  much greater than present day values.**

*48-51: Long sentence. Suggest to break it into two.*

Modified as follows:

A final experiment, esm-flat10-cdr, assesses climate reversibility under negative emissions, where we find that peak warming may occur before or after net zero and residual warming after removal of all greenhouse gases is well described by ZEC **in most models. Further, we find Simple Climate Models (SCMs)** tend to overestimate temperature reversibility compared with ESMs.

*58: The proportional relationship was also demonstrated with observational data – see Gillett et al., 2013.*

Good point. Modified as follows:

The relationship has its origins in the recognition of a robust linear relationship in Earth System Model simulations (Allen et al., 2009; Matthews et al., 2009; Zickfeld et al., 2009) **and observations (Gillett et al., 2013)** between the global mean temperature change and the cumulative amount of $CO_2$ released into the atmosphere, the slope of which we refer to as the Transient Climate Response to Cumulative Emissions (TCRE) - the change in global mean temperature per trillion tonnes of carbon emitted to the atmosphere.

*68-69: Several corrections must be applied to calculate emissions budgets using TCRE, not just for temperature impacts of non-CO2 emissions. These include human-induced warming, Earth system feedbacks among others (e.g. IPCC AR6 WGI, Ch. 5, Fig. 5.31).*

Thanks. We agree that this initial introduction needs to list all corrections to TCRE (they were previously mentioned later in the text). We've restructured the introduction a little to do this.

*80: TCR corresponds to the term DT(70)/I_atmos(70) (not DT(70)) in this equation.*

We disagree on this one. From the AR6 glossary, TCR is in units of K:

**TCR (Transient Climate Response): "The surface temperature response for the hypothetical scenario in which atmospheric carbon dioxide (CO2 ) increases at 1% yr $^{-1}$ from pre-industrial to the time of a doubling of atmospheric CO2 concentration (year 70)."**

*84: "deviations from cumulative emissions proportionality" manifest in different ways, not just as ZEC. They include deviations from linearity at higher levels of cumulative CO2 emissions (>1,500 GtC; IPCC AR6 WGI, Ch. 5) that are unrelated to ZEC or (ir)reversibility. Discuss these sources of deviation and clarify that ZEC is one of them or reword the sentence.*

Point taken -we've reworked this section.  It's mostly about ZEC here (in the context of carbon budgets for ambitious climate targets).

*87: A clearer definition of RAZE and its relationship to ZEC would be helpful.*

Agreed - we've expanded this discussion somewhat:

Alternative frameworks have been proposed to provide a less scenario-dependent formulation for ZEC. Consideration of linear pulse-response models of the climate show that cumulative emissions proportionality is an expected first-order response, but that second order terms allow for further temperature changes after emissions have ceased (Avakumović, 2024; Jenkins et al., 2022).  This second order behaviour can be approximated by 'Rate of Adjustment to Zero Emissions' or RAZE, which defines the fractional change in $CO_2$-induced warming after $CO_2$ emissions cease(Jenkins et al., 2022).  In this approximation (valid for decadal timescales following net zero), RAZE can be related to ZEC for a given scenario as:

$$ZEC_H = I_{em}(t = t_{net-zero})(TCRE)(RAZE)(H)|$$

where $ZEH_H$ is the warming $H$ years after net zero and $I_{em}(t{=}t_{net\text{-}zero})$ is the cumulative emissions at the time of net zero.

*111-112: "...inhibit a clear diagnosis of whether and how the general response to negative emissions differs from the climate response to positive emissions":*

*There are two issues with the CDRMIP 1pctCO2-cdr reversibility experiment:*

*i) the asymmetry in compatible CO2 emissions and removals, which is addressed in the flat10-cdr experiment,*

*ii) the lagged response of the coupled climate-carbon cycle system to positive emissions, which confounds the response to negative CO2 emissions. This issue is exacerbated in the 1pctCO2-cdr experiment due to the abrupt transition from large positive to large negative emissions, but is also present in the flat 10-cdr experiment. This should be acknowledged. Two ways have been proposed to isolate the response to negative emissions: subtract the committed response to positive emissions from the overall response (Zickfeld et al., 2016; Koven et al., 2023; Chimuka et al., 2023), or apply negative emissions from an equilibrium state (Zickfeld et al., 2021). Neither approach is entirely accurate.*

Good point - addressed as follows:

"Secondly, the lagged effects of the positive emission phase can complicate assessment of the response to negative emissions.(Chimuka et al., 2023; Koven et al., 2023; Zickfeld et al., 2016)

These confounding effects inhibit a clear diagnosis of whether and how the general climate response to negative emissions differs from the climate response to positive emissions (MacDougall, 2019). An idealized CMIP experiment that allows for a continuous transition from positive to negative emissions, and one that is symmetric in time (so that any asymmetries that arise are driven by the coupled carbon-climate response itself), improves on this status quo (though the separation of lagged effects remains a challenge)."

*124: "isolate the impact of global scale removals": How will this be done? Accurately isolating the response to removals is not possible with the proposed experimental design as argued above.*

Point taken (we will remove the word *isolate*) - though on this point, we would argue that this is a good reason to run the flat10-nz experiment, to assess the lagged effect of any previous positive emissions which might manifest after year 150 in flat10-cdr.

*198-200, "TR1000 would be a measure...": Unclear what TR1000 measures exactly based on this description. The warming centered around year 100 could itself deviate from TCRE nonlinearity, therefore subtracting "the expectation from TCRE" may not be the same as subtracting "a 21 year average around year 100".*

Agreed - reworded: "TR1000 would be a measure of hysteresis in global mean temperature when cumulative emissions return to 1000PgC on the downward branch minus the warming at the same cumulative emissions level under esm-flat10. "

*215: "This ensemble...": There appears a sentence missing that serves as transition from the experimental to the models that will be used to run the experiments.*

Added a new section header to make this clear

*216; "each model is selected for its specific configuration". Unclear – do you mean "categorized", e.g. into ESM, EMIC, SCM ?*

Sentence removed.

*276, "majority stabilize rapidly": Land carbon in most models appears to continue to decline slightly after year 100.*

Para rewritten as follows:

"Different models show a diversity of evolution of land and ocean carbon sinks - with some models (e.g. MPI, GFDL, GISS) initially absorbing land carbon for the first 20-50 years of the zero emissions phase before losing carbon on longer timescales,  while the land sink in other models (Uvic, GISS, HadCM3, UKESM) stabilise the land sink after emissions cessation.  Ocean uptake is more consistent across the ensemble, with all models simulating a continued uptake of carbon in the ocean during the zero emissions phase."

*279-281: Repeats experimental design description.*

deleted

*282, "... indicative on non-proportionality of temperature with cumulative emissions". This finding could be described in more detail as the temperature response at year 300 differs substantially from the temperature response in concentration driven simulations, with near zero warming or net cooling relative to year 0 in many models.*

Added the following at the start of 3.2.3:

"Global mean results for esm-flat10-cdr are shown in Figure 4.  Temperature response at year 300 (when cumulative emissions return to zero) show a range of -.7K to +.5K, indicating notable deviations from cumulative emissions proportionality with residual warming or cooling depending on the model. "

*290: This should probably read "timing of the PEAK land sink relative to net zero".*

Agree - done.

*310-311: "Similarly...": This sentence does not seem to belong here as the paragraph describes the carbon cycle response.*

Deleted.

*314: "Transient CLIMATE response to positive emissions".*

Modified as suggested.

*344-345: You could report ZEC 90 for FLATMIP simulations to allow a proper comparison with ZECMIP simulations.*

Agreed - done.

*350: T_1000PgC and T_100yr not defined.*

Added a line to introduce these in 2.1.1:

"As such, we refer to TCRE derived from esm-flat10 and 1pctCO2 as $T_{100yr}$ and $T_{1000PgC}$ respectively. "

*365-366: Smaller ZEC in experiments with lower emission rates up to the point of net zero was noted in MacDougall et al (2020) and attributed to the warming and carbon cycle response being closer to equilibrium.*

Well noted.  Added in 3.2.2:

"This is consistent with (MacDougall et al., 2020) who found smaller ZEC in experiments with lower emission rates up to the point of net zero, proposing that both warming and carbon cycle response being closer to equilibrium. "

*372-373: Sentence unclear.*

Reworded as follows:

"Some models (NorESM, CESM2, MPI, CNRM) in the ZECMIP experiment suggest an apparent short term warming pulse following cessation of emissions, which is less pronounced in the *esm-flat10-zec* experiment (Figure 9)"

*480-481: Statement unclear.*

Deleted.

*Fig. 2: The temperature trajectory does not correspond to the actual response in the flat10-cdr experiment. In emission driven simulations warming after all CO2 is removed from the atmosphere should be closer to zero.*

Adjusted to make this less extreme.

*Fig. 3: DGMST axis labels in panels b, c: Use either T_1000PgC or T100yr consistently (assuming they are the same here).*

Done

*Fig. 6: Circles could be shown in panel d) for comparison.*

Done - plotting error that they weren't showing up.

*Fig. 10: Horizontal axis labels not positioned properly.*

Fixed

*Fig. 12: Units missing in axis labels.*

Fixed, and ZEC90 added.

**flat10MIP: Response to Reviewer 2**

*This study analyses the flat10MIP experiments as a method to measure key coupled climate-carbon-cycle properties in Earth System Models, with the experimental protocol outlined in Sanderson et al. (2024). The authors frame the novelty of this work around the experimental design (CMIP models are forced with constant CO2 emissions at 10GtCO2/yr for a period of 100 years), meaning that models are run in emissions-driven configurations, forced at approximately present-day emissions rates and all react to the same quantity of cumulative CO2 emissions. This is a nice advancement – it is a more relevant perturbation for characterising key model response characteristics, such as the TCRE, ZEC and the extent of reversibility for CO2-induced warming.*

*This study estimates TCRE and ZEC from flat10 and flat10cdr experiments, finding TCRE is slightly lower on average (0.1K) than when measured with DECK 1%/yr CO2 concentration increase experiments. ZEC is shown to be 20-25% lower in this experiment configuration.*

*The manuscript is generally well written, although there are several places where greater discussion would be beneficial, particularly linking to processes which could be responsible for the emergent behaviours described in this work, and a handful of typos etc to be caught. Specific comments are left below. A general tightening of the written text and figure captions would be valuable before final submission.*

Many thanks for the positive assessment, we have endeavoured to tighten up the language and add key additional context as suggested by the reviewers.

***Specific comments***

*ZEC needs to be qualified over a timescale throughout the paper – ZEC is time dependent and thus should not be talked about as being e.g. 20-25% smaller, without qualifying the timescale over which this was measured.*

We've gone through and made explicit which ZEC we're referring to on each occasion.

*Line 50/51: "may be overestimate…" change to "may overestimate…"*

Corrected, thanks.

*Line 58: Observational constraints are also possible, although granted these do all require a model of some kind to determine characteristic warming response shapes, but not necessarily only ESMs.*

Good point - added reference to observational constraints:

"The relationship has its origins in the recognition of a robust linear relationship in Earth System Model simulations (Allen et al., 2009; Matthews et al., 2009; Zickfeld et al., 2009) and observations (Gillett et al., 2013) between the global mean temperature change"

*Line 86/89: RAZE could instead be defined as measure of the residual warming trend over a multidecade interval following net zero, normalised by the warming level. That way it is consistent with the definition of ZEC, where both then refer to their measuring of post-net-zero warming behaviours.*

We've added a consolidated paragraph on RAZE, which covers this point:

Alternative frameworks have been proposed to provide a less scenario-dependent formulation for ZEC. Consideration of linear pulse-response models of the climate show that cumulative emissions proportionality is an expected first-order response, but that second order terms allow for further temperature changes after emissions have ceased (Avakumović, 2024; Jenkins et al., 2022). This second order behaviour can be approximated by 'Rate of Adjustment to Zero Emissions' or RAZE, which defines the fractional change in $CO_2$-induced warming after $CO_2$ emissions cease(Jenkins et al., 2022). In this approximation (valid for decadal timescales following net zero), RAZE can be related to ZEC for a given scenario as:

$$ZEC_H = I_{em}(t = t_{net-zero})(TCRE)(RAZE)(H)$$

where $ZEH_H$ is the warming $H$ years after net zero and $I_{em}(t=t_{net-zero})$ is the cumulative emissions at the time of net zero. In this framing, a linear estimate of warming rate after net zero, if emissions are held at net-zero, is given by $I_{em}(t = t_{net-zero})(TCRE)(RAZE)$.

*Line 121/123: Again, clarity on the time horizon of this ZEC assessment is needed.*

 Agreed, added the following:

"·        Assess the Zero Emissions Commitment across models on multiple timescales: systematically measure the unrealized warming that continues after all $CO_2$ emissions have been halted (again, in an experiment where emissions are the independent variable), through assessment of ZEC after 50,90,100 and 200 years.

"

*Line 126/127: "Further studies are needed to"? or "Regional and component responses require further study beyond the scope of the globally aggregated analysis presented here"?*

 Agreed - we use your wording here:

"Regional and component responses require further study beyond the scope of the globally aggregated analysis presented here.  Studies in preparation will consider in detail commitment and reversibility of ocean heat transport, regional climatology and land carbon dynamics.

"

*Line 132/134:  suggest edit for readability? "Here, Flat10MIP simulates 3 of the 4 experiments proposed in (Sanderson et al., 2023) using CMIP6 generation models, as a pilot study in preparation for CMIP7."*

 Thanks - we use this edit as suggested.

*Line 149/152: Slight lack of clarity on what is requested, 150 or 300 yr simulations? I think you mean in this flat10MIP experiment set you requested 150 years, and in CMIP7 iteration models request could rise to 300 years, for reasons you lay out. Is that correct interpretation? Could it be clarified that over these experiments the expectation is to run +10GtC/yr emissions throughout?*

 Revised as follows:

"The protocol for *esm-flat10* is to continue emissions at 10PgC/year for the duration of the experiment.  150 years were conducted in this ensemble to allow the simulation to reach 2x pre-industrial $CO_2$ concentrations in most cases (allowing for a wide range of plausible land and ocean carbon uptake).  However, for future experiments in CMIP7 and beyond, a 300 year or longer *esm-flat10* would be useful to explore potential nonlinearities in response at higher cumulative emission levels which have been observed in some models(Schwinger et al., 2022).

"

*Line 157: Nice. Would be good to have this definition of ZEC earlier in paper, as per previous comments.*

 Thanks - moved up to first introduction of ZEC

*Line 166: "Cumulative net zero emissions" is a slightly unknown term. Could you say it without jargon? "...such that cumulative emissions return to zero, i.e. emissions and removals sum to zero."? Maybe not, but net zero is such a well understood term it may be worth reserving it for only describing annual emissions balance?*

 Fair point.  Rephrased:

"The esm-flat10-cdr experiment serves as an emissions-driven experiment to diagnose the response of the climate system to reducing, and ultimately reaching net-negative emissions and will provide a measure of climate reversibility when all cumulative anthropogenic emissions are removed (i.e. all cumulative emissions and removals sum to zero) at the end of the experiment."

*Line 174/180: This is good, glad to see this experiment also proposed. It would be good to briefly justify the expectation of a difference in response to a sudden cessation in emissions here. I.e. why do you need a gradual-NZ experiment as well as a sudden cessation one? Do you expect your ZEC measures in this study to have been impacted by the sudden cessation approach. Is such an experiment a useful measure of a true earth system property, or an emergent measure of a model's response to this particular scenario?*

 Thanks, we'd be interested in working more on the flat10-nz experiment.  Added the following:

"Such an experiment could help differentiate the response of the system to negative emissions in *esm-flat10-cdr* from the delayed response to positive emissions, and would provide a counterpoint to the abrupt emissions termination seen in *esm-flat10-zec* – providing an idealised scenario which might provide a more policy-relevant estimate of ZEC dynamics, reaching net-zero after a period emissions reduction.  "

*Line 216: potentially worth adding a qualifying sentence on the impact of measuring earth system properties from individual experiment runs with variability. Clearly ESMs will be impacted by this in a very different way to EMICs and SCMs.*

 Agree.  Added the following:

"We note that metrics from Earth System Models, unlike SCMs, are subject to uncertainty arising from internal variability. We would encourage centers to perform at least 3 members of these experiments in CMIP7 to provide better sampling and estimation of the role of initial condition uncertainty. "

*Line 224: closing bracket is missing.*

Thanks, corrected.

*Figure 2: I imagine this figure was thrown together quite quickly. Could it be tidied up a little? E.g. bring all text within figure bounds.*

Fair point...

[Figure]

*Line 242/244: Long sentence which I took a few read throughs to grasp. Potentially look to clarify.*

Split into 2 sentences:

"Figure 3 illustrates the global temperature response for the 3 simulations requested in flat10MIP. Throughout this section, we refer by default to $T_{100yr}$ the warming, in units K, after 1000PgC of cumulative emissions (which in *esm-flat10* occurs in year 100). $T_{100yr}$ is numerically equivalent to TCRE (units K/1000PgC) but allows proper consideration of the arithmetic sum with ZECn, also in units K. "

*Line 279: Some repetition here on the above experimental description. Suggest remove and refer to above.*

 Removed as suggested

*Line 283/285: Could also reference Jenkins et al. (2022) RAZE study. Negative RAZE corresponds to warming peaking before net zero. "as seen in similar experiments [Koven et al.] and ZECMIP experiments [Jenkins et al.]".*

 Agreed. Added.

*Line 303/311: What would be more informative is an attempt to link this explanation to the underlying mechanisms and possible processes which could cause this diversity of responses.*

 Agreed. Expanded as follows:

"Figure 5 shows how the rate of carbon emission allocation to the atmosphere, land, and ocean evolves as a function of time in the different experiments. In *esm-flat10*, we observe a transition from an initially high airborne fraction towards increasing allocation to land and ocean pools, with the airborne fraction in year 100 ranging between 0.45 and 0.55 across models. This variation arises from inter-model differences in the representation of land and ocean carbon uptake processes. For example, some models exhibit sustained terrestrial uptake (e.g., CESM2, NorESM2), while others (e.g., ACCESS, UKESM) show land sink saturation or reversal, likely reflecting the interplay between $CO_2$ fertilization(Arora et al., 2020), nutrient availability (Goll et al., 2012) and warming-induced soil carbon losses (MacDougall et al., 2020; Wieder et al., 2013). Declining land uptake in some models may also reflect increasing hydrological stress or climatic constraints on productivity(Fisher et al., 2019). During the *esm-flat10-zec* experiment, atmospheric $CO_2$ declines following cessation of emissions, but models diverge in whether this drawdown is primarily balanced by land (e.g., GFDL, CNRM) or ocean (e.g., GISS, ACCESS) uptake. These differences reflect the distinct timescales and sensitivities of the carbon pools: the land sink responds quickly to emissions cessation but may decay as $CO_2$ fertilization effects diminish and heterotrophic respiration increases(Jones et al., 2013), while the ocean continues to absorb carbon due to its longer equilibration timescales and sustained $pCO_2$ disequilibrium (Schwinger and Tjiputra, 2018; Tjiputra et al., 2013) and model-specific representation of deep ocean ventilation and carbon transport (Séférian et al., 2024). The resulting diversity in sink partitioning highlights key model-dependent feedbacks in the terrestrial biosphere and ocean circulation, which modulate the climate system's reversibility following net-zero.

"

*Figure 7: Colouring is not very readable. White text on light colours particularly.*

 Revised:

[Figure]

*Line 348/349: Is there an explanation for this that could be offered?*

Expanded a little:

"This is consistent with (Krasting et al., 2014), who found that TCRE estimated at high emissions rates was greater than that estimated using present day emissions rates and attributed the difference to a greater disequilibrium between land/atmosphere and ocean response states when emissions rates are very high. "

*Line 350/353: FaIR and MAGICC and CICERO-SCM are all tuneable to have various responses. What does the distribution of model responses actually tell us? How have the three parameter distributions been selected which you run over each model? Surely this is a key reason for an observed differences in model response distribution?*

Fully agreed.  Added the following:

"Given that probabilistic calibration is performed independently for each SCM, it is not easy to attribute these differences to structural differences between the models or to choices of probabilistic parameter calibration strategy."

*Line 364/373: Is it inevitable that the ZEC will be lower under this experiment design, compared to ZECMIP, since ZEC has demonstrated proportionality to the "average cumulative emissions over the period" [Jenkins et al 2022], which is different under each experiment design? This could be addressed as a point in the final paragraph of this section [lines 384-390].*

 Agreed - added the following:

"In contrast, behavior in *esm-flat10-zec* has slightly less warming during the positive emissions phase, and less adjustment afterwards, resulting in lower values for TCRE and smaller magnitudes (either positive or negative) of ZEC50 and ZEC90.  The finding that ZEC50/90 from *esm-flat10* is lower than ZECMIP estimates is consistent with the findings of (Jenkins et al., 2022), who found that ZEC is modulated by "average cumulative emissions over the period", a metric which is different under the two experimental designs."

*Line 412/417: This explanation of TCRE + ZEC is not a particularly precise explanation, given the ZEC is both time and scenario dependent. RAZE approach is an alternative (and more scenario and time dependent) justification of this emergent behaviour (at least from the perspective of a global impulse response model), and robustly identifies the emergent ZEC (multi-decadal) response as impacting before and during the net zero transition. In a symmetric experiment design like the one proposed in flat10cdr, one would then expect half of the ZEC to be realised at the time of net zero, in so far as we trust a FaIR-like IR model to be representative of the coupled climate-carbon-cycle response in ESMs.*

We agree - to an extent, but would hold out that there remains scope in the literature for a framework which can describe behaviour at a range of timescales.  We've added the following:

"Alternative frameworks such as RAZE (Jenkins et al., 2022), explain other key features – such as the expectation in a symmetrical experiment such as *esm-flat10-cdr* that half of the ZEC is manifested at the time of net zero.  A unifying explanation for these frameworks that is accurate both during the net zero transition and at timescales significantly before and after, remains absent from the literature to date."